# THE MEANING OF "MOST" FOR VISUAL QUESTION ANSWERING MODELS

## ABSTRACT

The correct interpretation of quantifier statements in the context of a visual scene requires non-trivial inference mechanisms. For the example of *"most"*, we discuss two strategies which rely on fundamentally different cognitive concepts. Our aim is to identify what strategy deep learning models for visual question answering learn when trained on such questions. To this end, we carefully design data to replicate experiments from psycholinguistics where the same question was investigated for humans. Focusing on the FiLM visual question answering model, our experiments indicate that a form of approximate number system emerges whose performance declines with more difficult scenes as predicted by Weber's law. Moreover, we identify confounding factors, like spatial arrangement of the scene, which impede the effectiveness of this system.

## 1    INTRODUCTION

Deep learning methods have been very successful in many natural language processing tasks, ranging from syntactic parsing to machine translation to image captioning. However, despite significantly raised performance scores on benchmark datasets, researchers increasingly worry about interpretability and indeed quality of model decisions. We see two distinct research endeavors here, one being more pragmatic, forward-oriented, and guided by the question *"Can a system solve this task?"*, the other being more analytic, reflective, and motivated by the question *"How does a system solve this task?"*. In other words, the former aspires to improve performance, while the latter aims to increase our understanding of deep learning models.

By 'understanding' here we mean observing a reasoning mechanism that, if not resembling human behavior, at least is cognitively plausible. This is by no means necessary for practically solving a task, however, we highlight two reasons why being able to explain model behavior is nonetheless important: On the one hand, cognitive plausibility increases confidence in the abilities of a system – one is generally more willing to rely on a reasonable than an incomprehensible mechanism. On the other hand, pointing out systematic shortcomings inspires systematic improvements and hence can guide progress. Moreover, particularly in the case of a human-centered domain like natural language, ultimately, some degree of comparability to human performance is indispensable.

In this paper we are inspired by experimental practice in psycholinguistics to shed light on the question of how deep learning models for visual question answering (VQA) learn to interpret statements involving the quantifier *"most"*. We follow Pietroski et al. (2009) in designing abstract visual scenes where we control the ratio of the objects quantified over and their spatial arrangement, to identify whether VQA models exhibit a preferred strategy of verifying whether *"most"* applies. Figure 1 illustrates how visual scenes can be configured to favor one over another mechanism.

We want to emphasize the experimental approach and its difference to mainstream machine learning practice. For different interpretation strategies, conditions are identified that should or should not affect their performance, and test instances are designed accordingly. By comparing the accuracy of subjects on various instance types, predictions about a subject's performance for these mechanisms can be verified and the most likely explanation identified. Note that our advocated evaluation methodology is entirely extrinsic and does not constrain the system in any way (like requiring attention maps) or require a specific framework (like being probabilistic).

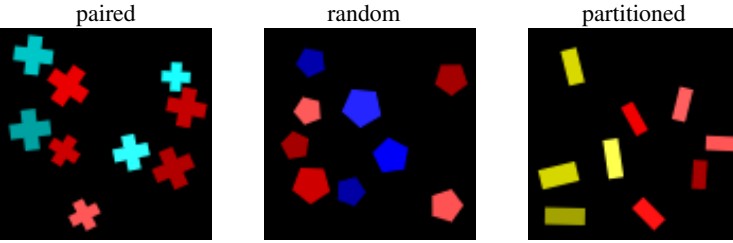

*"More than half the shapes are red shapes?"*

Figure 1: Three spatial arrangements of objects which may or may not affect the performance of a mechanism for interpreting *"most"* statements. Going from left to right, a strategy based on pairing entities of each set and identifying the remainder gets more difficult, while a strategy based on comparing cardinalities does not.

Psychology as a discipline has focused entirely on questions around how humans process situations and arrive at decisions, and consequently has the potential to inspire a lot of experiments (like ours) for investigating the same questions in the context of machine learning. Similar to psychology, we advocate the preference of an artificial experimentation environment which can be controlled in detail, over the importance of data originating from the real world, to arrive at more convincing and thus meaningful results.

Artificial data has a history in deep learning of establishing new techniques – most prominently, LSTMs were introduced by showing their ability to handle various formal grammars (Gers & Schmidhuber, 2001) – and our higher-level goal with this paper is to demonstrate the potential for more informative evaluation of machine learning models in general. This is motivated by our belief that, in the long term, true progress can only be made if we do not just rely on the narrative of neural networks *"learning to understand/solve"* a task, but can actually confirm our theories experimentally. Taking inspiration from psychology seems particularly appropriate in the context of powerful deep learning models, which recently are not infrequently described by anthropomorphizing words like *"understanding"*, and compared to *"human-level"* performance.

## 2    THE MEANING OF "MOST"

In this section we will introduce the two mechanisms of interpreting *"most"*, discuss cognitive differences and implications, and introduce relevant cognitive concepts.

### 2.1    GENERALIZED QUANTIFIERS AND "MOST"

*"Most"* has a special status in linguistics due to the fact that it is the most prominent example of a quantifier whose semantics cannot be expressed in first-order logic, while other simple natural language quantifiers like *"some"*, *"every"* or *"no"* directly correspond to the quantifier primitives $\exists$ and $\forall$ (plus logical operators $\wedge$, $\vee$ and $\neg$). This situation is not just a matter of introducing further appropriate primitives, but requires a fundamental extension of the logic system and its expressivity.

In the following, by $x$ we denote an entity, $\boldsymbol{A}$ and $\boldsymbol{B}$ denote predicates (*"square"*, *"red"*), $\boldsymbol{A}(x)$ is true if and only if $x$ satisfies $\boldsymbol{A}$, and $\mathbb{S}_{\boldsymbol{A}} = \{x : \boldsymbol{A}(x)\}$ is the corresponding set of entities satisfying this predicate (*"squares"*). Thus we can define the semantics of *"some"* and *"every"*:

$$\text{some/every}(\boldsymbol{A}, \boldsymbol{B}) \Leftrightarrow \exists/\forall x : \boldsymbol{A}(x) \Rightarrow \boldsymbol{B}(x) \tag{1}$$

Importantly, these definitions do not involve the concept of set cardinality and indeed can be formulated without involving sets. This is not possible for *"most"*, which is commonly defined in one of the following ways:

$$\text{most}(\boldsymbol{A}, \boldsymbol{B}) \Leftrightarrow |\mathbb{S}_{\boldsymbol{A} \wedge \boldsymbol{B}}| > 1/2 \cdot |\boldsymbol{A}|$$
$$\Leftrightarrow |\mathbb{S}_{\boldsymbol{A} \wedge \boldsymbol{B}}| > |\mathbb{S}_{\boldsymbol{A} \wedge \neg \boldsymbol{B}}| \tag{2}$$

This makes *"most"* an example of a **generalized quantifier**, and in fact all generalized quantifiers can be defined in terms of cardinalities, indicating the apparent importance of a cardinality concept to human cognition.

## 2.2 ALTERNATIVE CHARACTERIZATION

There is another way to define *"most"* which uses the fact that whether two sets are equinumerous can be determined without a concept of cardinality, based on the idea of a bijection:

$$A \leftrightarrow B :\Leftrightarrow \forall x : [A(x) \Leftrightarrow B(x)] \tag{3}$$

$$\Leftrightarrow |\mathbb{S}_A| = |\mathbb{S}_B| \tag{4}$$

The definition of equinumerosity can be generalized to *"more than"* (and, correspondingly, *"less than"*), which lets us define *"most"* as follows:

$$\text{most}(A, B) \Leftrightarrow \exists \mathbb{S} \subsetneq \mathbb{S}_{A \wedge B} : \mathbb{S} \leftrightarrow \mathbb{S}_{A \wedge \neg B} \tag{5}$$

Although, at a first glance, this definition looks similar to the one above, it can be seen as suggesting a different algorithmic approach to interpreting *"most"*, as we will discuss below.

## 2.3 TWO INTERPRETATION STRATEGIES

The two characterizations of *"most"* are of course truth-conditionally equivalent, that is, every situation in which one of them holds, the other holds, and vice versa. In particular, if we are just interested in solving a task involving *"most"* statements, we can be agnostic about which definition our system prefers. Nevertheless, the subtle differences between these two characterizations suggest different algorithmic mechanisms of verifying or falsifying such statements, meaning that a system processes a visual scene differently to come to the (same) conclusion about a statement's truth.

Characterization (2) represents the **cardinality-based strategy** of interpreting *"most"*:

1. Estimate the number of entities satisfying both predicates (*"red squares"*) and the number satisfying one predicate but not the other (*"non-red squares"*).
2. Compare these number estimates and check whether the former is greater than the latter.

We want to add that, actually, the two definitions in (2) already suggest a minor variation of this mechanism – see Hackl (2009) for a discussion on *"most"* versus *"more than half"*. However, we do not focus on this detail here, and assume the second variant in (2) to be 'strictly' simpler in the sense that both involve estimating and comparing cardinalities, but the first variant additionally involves the rather complex operation of halving one number estimate.

Characterization (5) utilizes the concept of a bijection, which is a comparatively simple pairing mechanism and as such could be imagined to be a primitive cognitive operation. This gives us the **pairing-based strategy** of interpreting *"most"*:

1. Successively match entities satisfying both predicates (*"red squares"*) uniquely with entities satisfying one predicate but not the other (*"non-red squares"*).
2. The remaining entities are all of one type, so pick one and check whether it is of the first type (*"red square"*).

## 2.4 COGNITIVE IMPLICATIONS

Finding evidence for one strategy over the other has substantial implications with respect to the 'cognitive abilities' of a neural network model. In particular, evidence for a cardinality-based processing of *"most"* suggests the existence of an **approximate number system** (ANS), which is able to simultaneously estimate the number of objects in two sets, and perform higher-level operations on the resulting number representations themselves, like the comparison operation here. Explicit counting would be an even more accurate mechanism here, but neither available to the subjects in the experiments of Pietroski et al. (2009) due to very short scene display time, nor likely to be learned by the 'one-glance' feed-forward-style neural network we evaluate in this work[1].

---

[1]By *"one-glance feed-forward-style networks"* we refer to the predominant type of network architecture which, by design, consists of a fixed sequence of computation steps before arriving at a decision. In particular, such models do not have the ability to interact with their input dynamically depending on the complexity of an instance, or perform more general recursive computations beyond the fixed recurrent modules built into their design. Important for the discussion here is the fact that precise – in contrast to approximate or subitizing-style – counting is by definition a recursive ability, thus impossible to learn for such models.

The ANS (see appendix in Lidz et al. (2011) for a summary) is an evolutionary comparatively old mechanism which is shared between many different species throughout the animal world. It emerges without explicit training and produces approximate representations of the number of objects of some type. They are approximate in the sense that their number judgment is not 'sharp', but resulting behavior exhibits variance – like interpreting *"most"* statements with a cardinality-based strategy, as described above. This variance follows **Weber's law** which states that the discriminability of two quantities is a function of their ratio[2]. The precision of the ANS is thus usually indicated by a characteristic value called **Weber fraction** which relates quantity and variance. The ANS of a typical adult human is often reported to have a Weber fraction of 1.14 or, more tangibly, it can distinguish a ratio of 7:8 with 75% accuracy. Finding evidence for the emergence of a similar system in deep neural networks indicates that these models can indeed learn and utilize more abstract concepts (approximate numbers) than mere superficial pattern matching (*"red squares"* etc).

Both mechanisms to interpret *"most"* suggest conditions in which they should perform well or badly. For the cardinality-based one, the difference in numbers of the two sets in question is expected to be essential: smaller differences, or greater numbers for the same absolute difference, require more accurate number estimations and hence make this comparison harder, according to Weber's law. The pairing-based mechanism, on the other hand, is likely affected by the spatial arrangement of the objects in question: if the objects are more clustered within one set, pairing them with objects from the other set becomes harder. Importantly, these conditions are orthogonal, so each mechanism should not substantially be affected by the other condition, respectively. By constructing (artificial) scenes where one of the conditions dominates the configuration, and measuring the accuracy of being able to correctly interpret propositions involving *"most"*, the expected difficulties can be confirmed (or refuted) and thus indicate which mechanism is actually at work.

Using this methodology, Pietroski et al. (2009) show that humans exhibit a default strategy of interpreting *"most"*, at least when only given 200ms to look at the scene and hence having to rely on an immediate subconscious judgment. This strategy is based on the approximate number system and the cardinality-based mechanism. Moreover, the behavior is shown to be sub-optimal in some situations where humans would, in principle, be able to perform better if deviating from their default strategy. Since machine learning models are trained by optimizing parameters for the task at hand, it is far from obvious whether they learn a similarly stable default mechanism, or instead follow a potentially superior adaptive strategy depending on the situation. While the latter is likely more efficient in solving at least a narrowly defined task, the former would instead suggest that the system is able to acquire and utilize core concepts like an approximate number system.

We may speculate about the innate preference of modern network architectures for either of the strategies: Most of the visual processing is based on convolutions which, being an inherently local computation, we assume would favor the pairing-based strategy via locally matching and 'cancelling out' entities of the two predicates. On the other hand, the tensors resulting from the sequence of convolution operations are globally fused into a final embedding vector, which in turn would support the more globally aggregating cardinality-based strategy. However, the type of computations and representations learned by deep neural networks are poorly understood, making such speculations fallacious. We thus emphasize again that the higher-level motivation for this paper is to demonstrate how we need not rely on such speculative 'narratives', but can experimentally substantiate our claims.

## 3 EXPERIMENTAL SETUP

The setup in this paper closely resembles the psychological experiments conducted by Pietroski et al. (2009), but aimed at a state-of-the-art VQA model and its interpretation of *"most"*.

---

[2]We want to emphasize that there is evidence for Weber's Law in a range of other approximate systems, some of them non-discrete and thus rendering a pairing-based strategy impossible. While this does not rule out such a strategy when observing performance decline as predicted by Weber's Law (which is probably not possible based on extrinsic evaluation alone), it strongly suggests that similar and thus non-pairing-based mechanisms are at work in all of these situations.

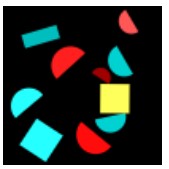
- Exactly two squares are yellow.
- Exactly no square is red.
- More than half the red shapes are squares.
- More than a third of the shapes are cyan.

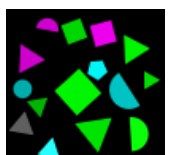
- Less than half the shapes are green.
- Exactly all magenta shapes are squares.
- At most five shapes are magenta.
- At least one triangle is gray.

Figure 2: Two example images with four in-/correct captions each, taken from the Q-full dataset (all quantifier and number captions).

### 3.1 TRAINING AND EVALUATION DATA

We use the ShapeWorld framework (Kuhnle & Copestake, 2017) as starting point to generate appropriate data. ShapeWorld is a configurable generation system for abstract, visually grounded language data. A data point consists of an image, an accompanying caption, and an agreement value indicating whether the caption is true given the image. The underlying task, image caption agreement, essentially corresponds to yes/no questions and as such is a type of visual question answering. Internally, the system samples an abstract world description from which a semantic caption representation is extracted. Both are then turned into 'natural' (but still abstract) representations as image and natural language statement, respectively. The latter transformation is based on a semantic grammar formalism (see their paper for details).

We use the pre-implemented quantifier captioner component, both in its unrestricted version and one with available quantifiers restricted to *"more than half"* and *"less than half"*[3]. The former contains various additional (generalized) quantifiers (*"no"*, *"a/three quarter(s)"*, *"a/two third(s)"*, *"all"*) and numbers (ranging from *"zero"* to *"five"*), each in combination with a comparing modifier (*"less than"*, *"at most"*, *"exactly"*, *"at least"*, *"more than"*, *"not"*). We refer to the unrestricted version as Q-full, the other one as Q-half. Figure 2 shows two images together with potential Q-full captions.

We also use the default world generator to produce training data (up to 15 randomly positioned objects, as seen in figure 2). However, all of the pre-implemented generator modules are too generic for our evaluation purposes, since they do not allow to control attributes and positioning of objects to the desired degree. We thus implemented our own custom generator module with the following functionality to produce test data.

**Attribute contrast:** For each instance, either the attribute 'shape' or 'color' is picked[4], and subsequently two values for this attribute and one value for the other is randomly chosen. This means that the only relevant difference between objects in every image is either one of two shape or color values (for instance, red vs blue squares, or red squares vs circles).

**Contrast ratios:** A list of valid ratios between the contrasted attributes can be specified, from which one will randomly be chosen per instance. For instance, a ratio of 2:3 means that there are 50% more objects with the second than the first attribute. We look at values close to 1:1, that is, 1:2, 2:3, 3:4, 4:5, etc. The increasing difficulty (for humans) resulting from closer ratios is illustrated in figure 3. Multiples of the smaller-valued ratios are also generated (e.g., 2:4 or 6:9), within the limit of up to 15 objects overall.

**Area-controlled:** If this option is set, object sizes are not chosen uniformly across the entire valid range, but size ranges for the two contrasting object types are adapted to the given contrast ratio and size of the chosen shape(s), so that both attributes cover the same image area on average. This means that the more numerous attribute will generally be represented by smaller objects, and the difference in covered area between, for instance, squares and triangles is taken into account.

---

[3] We use these two instead of *"most"* since ShapeWorld generates them by default. The VQA model is trained from scratch on this data, so we do not expect any of the differences between *"most"* and *"more than half"* one observes with humans to matter (Hackl, 2009).

[4] Note that we chose the examples in figures to always vary in color only.

Figure 3: From left to right, the ratio between the two attributes is increasingly balanced.

While objects are still positioned randomly in the basic version of this new generator module, we define two modes which control this aspect as well. Figure 1 in the introduction illustrates the different modes.

**Partitioned positioning:** An angle is randomly chosen for each image, and objects of the contrasting attributes are consistently placed either on one side or the other.

**Paired positioning:** If there are objects of the contrasted attribute which are not yet paired, one of them is randomly chosen and the new object is placed next to it.

The captions of these evaluation instances are always of the form *"More/less than half the shapes are X"*. with *"X"* being the attribute in question, for instance, *"squares"* or *"red"*. Note that this is an even more constrained captioner than the one used for Q-half, since the subject is always fully underspecified as *"shape"*. We also emphasize that, in contrast to this new evaluation generator module, the default generator configuration of the 'quantification' dataset pre-specified in Shape-World is used to generate the training instances in Q-half and Q-full. So these images generally contain many more than just two contrasted attributes, and ratios between attributes tend to be accordingly smaller. The examples in figure 2 are chosen to illustrate this fact: the second example contains a *"half"* statement with ratio 7:8, and the first contains one about a 0:4 ratio, while the image would also allow for a more 'interesting' 3:4 ratio (color of semicircles).

While we generally try to stay close to the experimental setup of Pietroski et al. (2009), in the following we point out some differences. Most importantly, instead of just using yellow and blue dots, we use all eight shapes and seven colors that ShapeWorld provides. This increases the visual variety of the instances and thus encourages the system to actually learn the fact that shape and color are attributes that can be combined in any way, instead of just straightforward binary pattern matching. Note that the humans in the psychological experiments have learned language in even more complex situations, which we cannot hope to approximate here. Moreover, our data does not contain yes/no questions but true/false captions, and *"most"*-equivalent phrasings *"more/less than half"*. Since the model is trained from scratch on such data, this should not affect results.

We do not implement their 'column pairs mixed/sorted' modes since they would require comparatively big and mostly empty images, hence require bigger networks and might cause practical learning problems due to sparseness, which we do not want to address here. In contrast, our 'partitioned' mode is more difficult than the ones investigated by Pietroski et al. (2009), at least for a pairing-based mechanism.

We will publish the generator configurations and custom generator modules required to reproduce the datasets we used here on acceptance of the paper.

## 3.2 MODEL

We focus on the FiLM model (Perez et al., 2018) here since it exhibited close-to-perfect accuracy on the CLEVR dataset (Johnson et al., 2017a), a diagnostic dataset for VQA which also consists of abstract images. We interpret the ShapeWorld captions and agreement values as questions and answer, respectively. The image is processed using either a pre-trained CNN or a four-layer CNN trained from scratch on the task. The question is processed by a GRU. In a sequence of four residual blocks, the image information is processed with its features linearly modulated (scale, offset) conditioned on the processed question embedding. Finally, the classifier module produces the answer, true or false. We use the code made available by the authors of the FiLM model, without changing any parameters. The only aspect we adapt is the trainable four-layer CNN, which uses a kernel size of 3, batch normalization and a stride of 2 in the second and fourth layer.

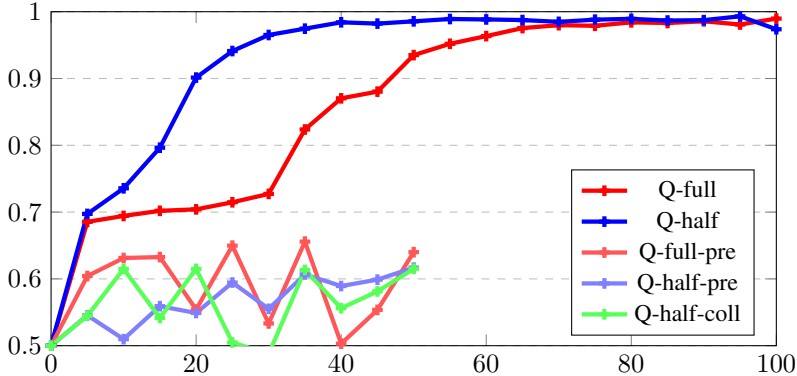

Figure 4: Training performance (iterations in 1000).

We considered investigating other models as well: The PG+EE model (Johnson et al., 2017b) is openly available and achieved very good performance on CLEVR, however, it relies on the 'program tree' provided by CLEVR, and while there exists a basic conversion of ShapeWorld caption models to CLEVR program trees, first, the CLEVR-specific modules do not cover quantifiers like *"most"* and, second, these program trees encode the interpretation strategy, which would defeat the purpose of our investigation to analyze precisely this mechanism as learned from data. The RelationNet architecture (Santoro et al., 2017) explicitly implements a pairing-based mechanism and hence we considered its evaluation less interesting than FiLM. For similar reasons, we did not focus on the VQA model of Zhang et al. (2018), whose architecture includes an explicit counting component. While our aim is to investigate the strategy for understanding *"most"* learned from data, it would be interesting to examine in both cases whether their architectural prior does indeed have the expected effect. Finally, we only learned about the MAC model (Hudson & Manning, 2018) after we started this project and so decided to leave it for future work, but we definitely consider it one of the most interesting candidate models to evaluate, since its architecture does not suggest an obvious preference for either strategy.

### 3.3 TRAINING DETAILS

The training set for both Q-full and Q-half consists of around 100k (25x 4096) images with 5 captions per image, so overall around 500k instances. The model is trained for 100k iterations with a batch size of 64. Training performance is measured on an additional validation set of 20k instances. Moreover, we produced 1024 instances for each of the overall 48 evaluation configurations, to investigate the trained model in more detail.

## 4 RESULTS

**Training.** We train two versions of the FiLM model, with CNN trained from scratch on the task: one on the Q-full dataset which contains all available quantifier and number caption types, the other on the Q-half dataset which is restricted to captions involving the quantifier *"half"* only. Performance of the system over the course of the 100k training iterations is shown in figure 4. The two models, referred to by Q-full and Q-half below, learn to solve the task quasi-perfectly, with a final accuracy of 98.9% and 99.4% respectively. Not surprisingly, the system trained on the more diverse Q-full training set takes longer to reach this level of performance, but nevertheless plateaus after around 70k iterations.

For the sake of completeness, we also include the performance of other models in this figure, which failed to show clear improvement over the first 50k iterations. This includes the FiLM model with pre-trained instead of trainable CNN module (Q-full-pre, Q-half-pre), and an earlier trial on Q-half (Q-half-coll) where we did not constrain the data generation to not produce object collisions (the default in ShapeWorld is to allow up to 25% area overlap). We note, however, that we have not done any hyperparameter search which might alleviate these learning problems.

Figure 5: Accuracy (in %) of model trained on Q-full and Q-half for the various evaluation setups.

| train | mode | size-controlled | | | | | | | | area-controlled | | | | | | | |
|---|---|---|---|---|---|---|---|---|---|---|---|---|---|---|---|---|---|
| | | all | 1:2 | 2:3 | 3:4 | 4:5 | 5:6 | 6:7 | 7:8 | all | 1:2 | 2:3 | 3:4 | 4:5 | 5:6 | 6:7 | 7:8 |
| Q-full | random | 92 | 100 | 99 | 97 | 94 | 91 | 88 | 85 | 93 | 100 | 99 | 97 | 93 | 91 | 86 | 82 |
| | paired | 93 | 99 | 99 | 96 | 93 | 90 | 88 | 82 | 93 | 99 | 99 | 96 | 91 | 87 | 84 | 80 |
| | part. | 89 | 100 | 99 | 92 | 90 | 81 | 77 | 72 | 89 | 99 | 98 | 92 | 88 | 82 | 78 | 72 |
| Q-half | random | 92 | 100 | 100 | 98 | 93 | 88 | 88 | 87 | 93 | 100 | 100 | 97 | 92 | 86 | 85 | 82 |
| | paired | 92 | 100 | 100 | 96 | 90 | 86 | 84 | 79 | 92 | 100 | 99 | 96 | 87 | 84 | 79 | 76 |
| | part. | 91 | 100 | 99 | 96 | 86 | 83 | 83 | 80 | 91 | 100 | 99 | 94 | 89 | 83 | 83 | 80 |

**Evaluation.** Table 5 presents a detailed breakdown of system performance on the evaluation settings. Before discussing the results in detail, we want to reiterate three key differences between the evaluation data and the training data:

- The visual scenes here do all exhibit close-to-balanced contrast ratios, while this is not the case for the training instances.
- The evaluation scenes only contain objects of two different attribute pairs, and consequently the numbers to compare are generally greater than in the training instances, where more attributes are likely present in a scene.
- Q-full contains not just statements involving *"half"* – in fact, a random sample of 100 images / 500 captions suggests that they constitute only around 8% of the dataset (and this includes combinations with modifiers beyond *"more/less than"*).

Considering that, the relatively high accuracy on test instances throughout indicates a remarkable degree of generalization.

**More balanced ratios.** The most consistent effect is that more balanced ratios of contrasted attributes cause performance to decrease. This is certainly affected by the tendency of the training data to not include many examples of almost balanced ratios. However, if this were the only reason, one would expect a much more sudden and less uniformly linear decrease. More importantly, since Q-full generally contains fewer *"half"* statements, the decline should be more pronounced here. We do not observe either of these effects, and thus conclude that both models may actually have developed an approximate number system. This is further discussed at the end of this section.

**Random vs paired vs partitioned.** There is definitely a clear negative effect of the partitioned configuration on performance for the model trained on Q-full, which indicates that the learned mechanism is not robust to a high degree of per-attribute clustering. This does not indicate a preference for the pairing-based strategy, though, since both models perform best on the random configuration. While this suggests that there is neither a preference for the perfectly clustered partitioned nor for the perfectly mixed paired arrangement, we note that the effect is not strong, and that these instances are most similar to the random placement of objects in the training data, which might cause this effect.

**Size- vs area-controlled.** The performance in both cases is comparable, showing that the models do not (solely) learn to rely on comparing the overall covered area, which would only work well in the size-controlled mode. Nevertheless, we note a tendency for area-controlled instances to be somewhat more difficult in random and paired mode, more so for Q-half, which suggests that the model(s) learn to use covered area as a feature to inform a correct decision in some cases.

**Q-full vs Q-half.** There seems to be a tendency of the system trained on Q-full to perform marginally better, except for the partitioned mode discussed before. The fact that this model performs at least on a par with the one trained on Q-half, while only seeing a fraction of directly relevant training captions, indicates that the learning process is not 'distracted' by the variety of captions, and indeed might profit from it.

**Ratios and Weber fraction.** We generated evaluation sets of even more balanced ratios (8:9, 9:10, 10:11, increasing the overall number of objects accordingly to 17/19/21), and in figure 6 plotted the accuracy of the Q-full model on increasingly balanced sets for all three spatial configuration modes,

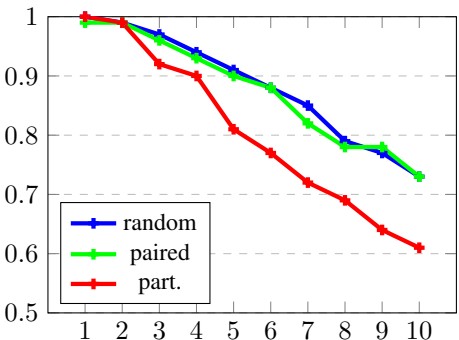 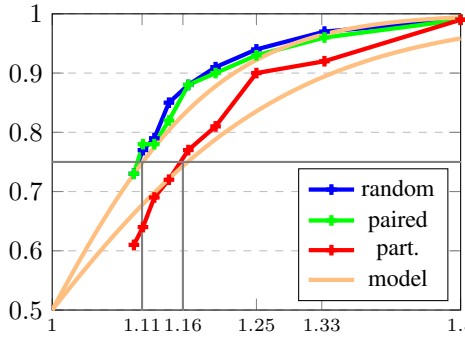

Figure 6: *(Left)* Q-full model performance for increasingly balanced ratios (x-axis: ratio as n:n+1). *(Right)* Performance as a function of actual ratio fraction n+1/n, with Weber fractions (75%) highlighted and the corresponding idealized model Weber curves indicated.

not controlling for area (which for greater numbers only has a negligible effect anyway). The figure also contains a diagram with accuracy plotted against ratio fraction, which is more common in the context of Weber's law. The characteristic Weber fraction can be read off directly as the ratio at which a subject is able to distinguish two values with 75% accuracy. We observe around 1.11 for random/paired and 1.16 for partitioned, which corresponds to 9:10 and 6:7 as closest integer ratios. These values are in the same region as the average human Weber fraction, which is often reported as being 1.14, or 7:8.

We emphasize that these curves align well with the trend predicted by Weber's law, even for the ratios with more than 15 objects overall, where such situations have never been encountered during training. All this strongly suggests that the model learns a mechanism similar to an ANS, which is able to produce representations that can (at least) be utilized for identifying the more numerous set. It can in particular be concluded that the system does not actually learn to explicitly count, since we would then not expect to observe such fuzziness characteristic to an ANS.

Moreover, since performance is affected somewhat by the partitioned and the area-controlled modes, the interpretation of *"most"* seems to be informed by other features as well. As we noted earlier, since the model is trained to optimize this task, an adaptive strategy is not unexpected. On the contrary, more surprising is the fact that an ANS-like system emerges as a dominating 'backbone' mechanism, with additional factors acting as less influential 'secondary' features.

## 5 RELATED WORK

Visual question answering (VQA) is the general task of answering questions about visual scenes. Since the introduction of the VQA Dataset (Antol et al., 2015), this dataset was widely used as evaluation benchmark for multimodal deep learning. It provides a shallow categorization of questions, including basic count questions, however, these categories are far too coarse for our purposes.

Motivated by various problems with the VQA Dataset (Goyal et al., 2017; Agrawal et al., 2016), a range of artificial abstract datasets have been introduced recently. CLEVR (Johnson et al., 2017a) consists of rendered images of geometric objects and questions generated based on templates, covering some abilities like number or attribute comparison in more detail, but still in a fixed categorization. NLVR (Suhr et al., 2017) contains crowdsourced statements about abstract images, but does not sort them according to some criteria. Recently, the COG dataset (Yang et al., 2018) was introduced, which most explicitly focuses on replicating psychological experiments for deep learning models, hence most related to our work. However, their dataset does not contain any number or quantifier statements.

There is some work on investigating deep neural networks which look at numerosity from a more psychologically inspired viewpoint. Stoianov & Zorzi (2012) find that visual numerosity emerges from unsupervised learning on abstract image data. Zhang et al. (2015) look at salient object subitizing in real-world images, formulated as a classification task over five classes ranging from '0'

to '4 or more'. In a more general number-per-category classification setup, Chattopadhyay et al. (2017) investigate different methods of obtaining counts per object category, one of them inspired by subitizing. Moving beyond explicit number classification, Zhang et al. (2018) recently introduced a dedicated counting module for visual question answering.

Other work looks at a similar classification task, but for proper quantifiers like "no", "few", "most", "all", first on abstract images of circles (Sorodoc et al., 2016), then on natural scenes (Sorodoc et al., 2018). Recently, Pezzelle et al. (2018) investigated a hierarchy of quantifier-related classification abilities, from comparatives via quantifiers like the ones above to fine-grained proportions. Wu et al. (2018), besides investigating precise numerosity via number classification as above, also look at approximate numerosity as binary greater/smaller decision, which closely corresponds to our experiments. However, their focus is on the subitizing ability, not the approximate number system, and their experiments follow a different methodology in that they already train models on specifically designed datasets, while we deliberately leverage such targeted data only for evaluation.

On a methodological level, our proposal of inspiring experimental setup and evaluation practice for deep learning by cognitive psychology is in line with that of Ritter et al. (2017) and their shape bias investigation for modern vision architectures.

## 6 CONCLUSION

We identify two strategies of algorithmically interpreting *"most"* in a visual context, with different implications on cognitive concepts. Following experimental practice of similar investigations with humans in psycholinguistics, we design experiments and data to shed light on the question whether the state-of-the-art FiLM VQA model shows preference for one strategy over the other. Performance on various specifically designed instances does indeed indicate that a form of approximate number system is learned, which generalizes to more difficult scenes as predicted by Weber's law. The results further suggest that additional features influence the interpretation process, which are affected by the spatial arrangement and relative size of objects in a scene. There are many opportunities for future work from here, from strengthening the finding of an approximate number system and further analyzing confounding factors, to investigating the relation to more explicit counting tasks, to extending the evaluation to other visual question answering models which also exhibit good performance on related tasks (Hudson & Manning, 2018; Zhang et al., 2018; Santoro et al., 2017).

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
