# OpenReview forum: "The meaning of "most" for visual question answering models"
_ICLR.cc/2019/Conference_

### Official Review · AnonReviewer3 · 2018-10-21
**Interesting analysis of quantifier interpretation in a VQA model, but the theoretical discussion is unsatisfying**

**Rating:** 5
**Confidence:** 4

**Review:**

The paper analyzes the strategy that a visual question answering model (FiLM) uses to verify statements containing the quantifier "most" ("most of the dots are red"). It finds that the model is sensitive to the ratio of objects that satisfy the predicate (that are red) to objects that do not; as the ratio decreases (e.g. 10 red dots compared to 9 blue dots), the model's performance decreases too. This is consistent with human behavior.

Strengths:
* The introduction lays out an ambitious program of comparing humans to deep neural networks.
* The experimental results are interesting (although of modest scope) and support the hypothesis that the network is not counting the objects but rather is using an approximation that is sensitive to the ratio between the red and non-red items.

Weaknesses:
* The architecture of the particular model is described very briefly, and at multiple points there’s an implication that this is an investigation of “deep learning models” more generally, even though those models may vary widely. While the authors are using an existing model, they shouldn't assume that the reader has read the paper describing that model. I would like to see more discussion of whether it is at all plausible for this model to acquire the pairing strategy, compared to alternative VQA models (e.g., using relation networks).
* I found it difficult to follow the theoretical motivation for performing the work. The goal seems to be to test whether the network is performing the task in way that "if not human-like, at least is cognitively plausible". I don't understand what is meant by cognitively plausible but not human-like; perhaps an example of a cognitively implausible mechanism would help clarify this issue. Later in the same paragraph, the authors argue that "in the case of a human-centered domain like natural language, ultimately, some degree of comparability to human performance is indispensable". This assertion is not justified, and seems surprising to me; we have very useful natural language processing systems that do not perform in a way that is comparable to humans (the hedge "some degree of" is really neither here nor there). In general, I don't understand why we would want a visual question answering system that returns approximate answers -- isn't it better to have it count exactly how many red dots there are compared to non-red dots?
* The authors assume that explicit counting is not "likely to be learned by the 'one-glance' feed-forward-style neural network" evaluated in the paper. What is this statement based on? Why would a "one-glance" network have trouble counting objects? (What is a “one-glance network”?)
* Another vague concept that is used without clarification: it is argued that if the network implements something like the Approximate Number System, that shows that it can "learn and utilize higher-level concepts than mere pattern matching". What is "pattern matching" and how does it differ from "higher-level concepts"?
* Why would the pairing strategy in a neural network be affected by the clustering of the objects? I understand why a human who needs to saccade back and forth between the two groups of objects might lose track of the objects that have been paired so far, but I don't understand why that would affect the architecture in question.

Minor comments:
* Is the definition of "most" really a central piece of evidence for "the apparent importance of a cardinality concept to human cognition"? Our ability to count seems sufficient to me. Perhaps I'm not understanding what the authors have in mind here.
* Please use the terms "interpretation" and "verification" consistently.
* "One over the other strategy" -> "one strategy over the other".
* The paper is almost 9 pages long, but the contribution does not appear more substantial than a standard 8-page submission.

---

> ### Author Response · Authors · 2018-11-16
> **Authors' response (part 1)**
>
> Many thanks for the valuable feedback! We uploaded a revised version of the paper, and in the following address the weaknesses you pointed out:
>
> - We added a few sentences to the end of section 2.4 on our speculative intuition regarding what strategy a model may prefer, and we do indeed think that a pairing-based strategy is plausible for convolution-based networks. When talking in more general terms about "deep learning models", we refer to the proposed methodology for "investigating deep learning models", and don't want to claim that we actually evaluate a representative number of models. We see this methodology, as illustrated by our experiments for one model, as the central part of our contribution. The first paragraph of section 3.2 describes this FiLM model and, given the focus on methodology, we considered the description (plus reference to the paper) sufficient here.
>
> - There are a few points here:
> * Since it was shown that humans seem to follow a cardinality-based strategy, the pairing-based one would be not human-like, but nonetheless cognitively plausible. We use "cognitively" in the sense of "algorithmically" plausible, so a procedure that makes sense for solving the problem. An example for an implausible method would be to rely on color/shape cues to solve instances involving "most", which doesn't make sense for the abstract meaning of "most".
> * Regarding the question whether comparability to human behavior is indispensable: On the one hand, acceptance of and reliance on systems which follow vastly different principles can be problematic; on the other hand, if the information a system uses to arrive at its conclusion doesn't make any sense to humans (in cases where humans have an intuition what is relevant, like the above example of "most" and color/shape cues), we doubt that good performance alone will justify using such a model, as opposed to instead doubting the quality of the underlying benchmark data.
> * The question whether we want a VQA system which returns approximate answers is an interesting one, but we don't intend to claim that this is a desired property, just that it is desirable to know whether (and how exactly) our systems currently solve such tasks approximately. A conclusion from our findings may well be that it is worth improving VQA models with respect to their counting capability, as they seem to rely on an approximate as opposed to an exact number system.
>
> - We added a longer footnote to section 2.4 about the "one-glance feed-forward-style networks" for clarification. In summary, general precise counting is an inherently recursive ability, and models which don't have an architectural module for recursive computations are not expected to be able to learn this capability, while they may stil learn to subitize or represent numbers approximately (which doesn't require recursion).
>
> - You're right, this sentence was a bit vague, we rephrased it to: "these models can indeed learn and utilize more abstract concepts (approximate numbers) than mere superficial pattern matching ("red square" etc)". The differences we want to point out is that, on the one hand, (approximate) numbers are a more abstract concept than, for instance, object types like "cat", "chair", etc as they can be combined with any object type. On the other hand, being able to utilize such representations to answer practical questions like whether "most" applies is more interesting than just being able to classify which representation applies.
>
> - Good point, our reasoning here was mostly influenced by Pietroski et al.'s work and our intuition about the ability of convolutions to learn a local pairing strategy (see addition at the end of section 2.4). Presumably, it could be possible to observe the behavior in our paper based on a pairing-based mechanism which works approximately, independent of clustering, as predicted by Weber's Law. It's probably impossible to ultimately rule out a pairing-based strategy via experiments evaluating extrinsic behavior only, but we note that there is evidence for Weber's Law in other approximate systems where pairing-based strategies are no alternative, thus suggesting that similar mechanisms are at work here. We added a footnote on this to section 2.4.

---

> > ### Author Response · Authors · 2018-11-16
> > **Authors' response (part 2)**
> >
> > - There are a few papers focusing solely on "most" in psycholinguistics (like the ones cited) and linguistics in general (e.g., formal semantics), many of which talk about cardinality as a "core concept" of human cognition, and many of which contrast a more subconscious concept of cardinality (like the approximate number system) with the conscious algorithmic ability for precise and infinite counting.
> >
> > - We consistently use "interpretation" in the new version.
> >
> > - Fixed.
> >
> > - We think that our approach and particularly its inspiration by experiments from psychology are a substantial contribution to evaluation methodology in the context of powerful deep learning models, which are not infrequently described by anthropomorphizing words like "understand" and compared to "human-level" performance (added a sentence to the introduction). The reason for exceeding 8 pages in length is likely due to the more elaborate introduction of the various concepts from Pietroski et al.'s work and experimental methodology in psychology in general, which we assume the audience of this conference is not very familiar with.

---

### Official Review · AnonReviewer1 · 2018-11-02
**Interesting direction and discussion to study the relationship “most” with limited experimental evaluation focusing on a single model.**

**Rating:** 5
**Confidence:** 5

**Review:**

Problem and contribution:
The paper studies if the Visual Question answering model “FILM” from Perez et al (2018) is able to decide if “most” of the objects have a certain attribute or color.
For this it tries to mimic the setup used to test human abilities in the study by Pietroski et al. (2009).

The main contribution of this is work is a discussion of how a model could solve the problem of deciding “most” and the study which shows that the studied model has some ability to do this. From this the paper concludes that the model is likely to have some approximate number system.


Strengths:
1.	The paper looks at a new angle to study and characterize CNN models in general, and VQA models in particular by looking into the psycholinguistic literature experimental setup studied with human subjects.
2.	The paper studies different variants of controlling for different factors (e.g. pairing data points, area used, different training data and pre-trained vs. trained from scratch CNN models)
3.	It is interesting to see that the models performance reasonably aligns with the curve predicted by “Weber’s law”.


Weaknesses:
4.	Number of objects vs. ratios is not disentangled: While the paper clarifies that not only a smaller number of objects are used, it would be interesting to understand if similar conclusions hold if only the same number or about the same number of total objects are used but the ratios change (at least for more extreme ratios, 1:2, this seems to be the case as they achieve 100% accuracy).
5.	The paper only focusses on a single VQA model (FILM) which limits the understanding if this observation is specific to this model; what about other models such as the one from Hudson & Manning (2018), or Relation Networks (Santoro et al) or even simpler baselines: A system which two attention mechanisms (without normalizations) which are sum pooled and then compared would sort of explicitly encode the idea of the APN system. It would be valuable to compare them to see how different systems (can) solve this task. I would expect that the architecture favors certain capabilities; e.g. Relation Networks might lead more to a paring-based strategy. Or Zhang et al. (2018) might be able to exploit explicit counting to solve the task.
6.	The “most” ability or APN ability seems to be highly related to accumulation in neural networks. The paper FiLM uses global max-pooling and I am wondering if this affect this ability.
7.	The study is only performed on symbols which a very large training set (given the difficulty of the problem) and it not clear how well this generalizes to real images or scenarios with less training data.
7.1.	Maybe beyond the scope of this work, but it would be interesting to understand how much training data different models need to obtain this capability.
8.	For evaluation: Are there distractors, i.e. elements which don’t belong to set A or B? If not, how would distractors affect it.
9.	Clarity:
9.1.	The equation between equation (1) and (2) misses a number [I will call it 1.5 for now]
9.2.	In formula (1.5) “<=>” seems to be used at different levels (?) it would be good to use brackets to make clear which level “<=>” refers to.

Minor:
10.	The title suggests that the paper studies multiple VQA models but only a single model is studied.

Conclusion:
The paper looks into an interesting direction to study CNN models but has some limitations including studying only a single VQA model type, limited to artificially generated images.

---

> ### Author Response · Authors · 2018-11-16
> **Authors' response**
>
> Many thanks for the valuable feedback! We uploaded a revised version of the paper, and in the following address the weaknesses you pointed out:
>
> 4. Note that the training data is not constrained with respect to ratios and number of objects addressed by the caption, so the learned behavior should be independent of these aspects. Moreover, note that for most ratios there is only one combination of numbers with at most 15 objects in total, but larger images fitting a greater total number of objects would definitely be an option here. For the less close-to-balanced ratios 1:2, 2:3, 3:4 where there are multiple possibilities, performance generally is (close-to-)perfect, indicating that there is no increased difficulty of learning multiples in the presence of more close-to-balanced ratios (for instance, 6:9 vs 7:8). We hope this clarifies your concern.
>
> 5. We fully agree that it would be very interesting to investigate these models. For this paper, we decided to focus on the methodology of investigating such questions in detail (the evaluation for FiLM alone comprises around 100 experiments) as opposed to focusing on the comparison of behavior of different models, and leave the latter to future work. We added a few additional sentences to section 3.2 regarding that.
>
> 6. We added a few sentences to the end of section 2.4 on our speculative intuition regarding what strategy a model may prefer. We didn't think about the fact that one may want to control which strategy is learned, which would indeed be interesting, but that's why we considered FiLM as is and didn't experiment with changing architecture details. At the same time, considering that understanding "most" is only one of many capabilities a VQA model is supposed to learn, these results are unlikely to be an important influencing factor for architecture choice, while at least knowing about the properties of a model is nonetheless interesting.
>
> 7. The evaluation is supposed to show what an architecture is capable of learning under "ideal" conditions. It's an interesting question whether/how this changes when gradually shifting towards "less ideal" setups. An advantage of using a controlled setup like ours is that this is possible to investigate, to some degree at least (for instance, add more types of captions to the training data, not just quantifier statements). At some point we may be interested in actually investigating the same for real-world data, but we think it's unclear right now what exactly such evaluation data should ideally look like, what problems are most interesting, what details to pay attention to. Artificial data allows us to investigate these questions while avoiding the elaborate and expensive process of obtaining real-world data.
>
> 8. Note that the training data is far less constrained than the evaluation data, including various distracting aspects like additional shapes/colors. The evaluation data doesn't contain such distractors, but it would of course be possible (and potentially interesting) to add such. We didn't do so since we considered instances with only "relevant" attributes to be the most difficult setup, like a minimal pair, where a model is required to focus on all objects and both their shape and color attribute to decide correctly.
>
> 9. We incorporated the changes as you suggested. Thanks!
>
> 10. We didn't think about this interpretation -- our intention was to signal that we take the "The Meaning of 'Most'" setup and methodology of Pietroski et al. from psychology, and implement a deep learning version for visual question answering models.

---

> > ### Comment · AnonReviewer1 · 2018-12-11
> > **Still Borderline**
> >
> > On the positive side:
> > + The paper improved in the revision, improving mainly discussion and increasing clarity.
> >
> > Remaining weaknesses:
> > - I still think for an analysis paper it is important to have a comparison of more than a single model. Even when proposing a new model we expect papers to compare to prior works, which might mean running them on new data; when proposing a new evaluation/study methodology I think is even more important to have an understanding of multiple methods. (The argument that "FiLM alone comprises around 100 experiments" is not a strong argument, I expect the experiments to be reasonable fast and other methods could be run just for the most important experiment/setting, i.e. training it once or twice)
> >
> > - The paper's conclusion remain limited due to the synthetic nature of the data.
> >
> > - R3 brought up the point of "one-glance feed-forward-style networks". The authors state that "precise counting is an inherently recursive ability". While this might be true for counting in general, for counting of small number, as e.g. studied in this work, the work of Zhang et al. (2018) shows an approach to do so with "one-glance feed-forward-style networks". Another reason to have more comprehensive evaluation of prior work.
> >
> > Overall I am not strongly opposed to accepting the paper, but I also think the study is limited and I remain with my border line rating of 5.

---

### Official Review · AnonReviewer2 · 2018-11-03
**Strong, hyper-focused contribution to VQA understanding**

**Rating:** 7
**Confidence:** 4

**Review:**

This paper studies how the FiLM visual question answering (VQA) model answer questions involving the quantifier ‘most’. This quantifier is chosen for study because it cannot be expressed in first order logic (i.e., high-order logic is required), and secondly because there are two different algorithmic approaches to answering questions involving ‘most’ (cardinality-based strategy and pairing-based strategy). Experiments are performed by designing abstract visual scenes with controlled numerosity and spatial layouts, and applying methodologies from pyscholinguistics. The paper concludes that the model learns an approximate number system (ANS), consistent with the cardinality-based strategy, with implications for understanding the conditions under which existing VQA models should perform well or badly (and possibly for improving VQA models).

Strengths:
- The research question is clear and well-conceived. In general, it seems there are significant opportunities for better collaboration between the experimental psychology and machine learning communities, and this is a good example of the benefits.
- The paper is clear, highly-focused, and well-written.

Weaknesses:
- The arguments for why the experimental evidence actually supports the existance of an approximate number system (ANS) could be made more clear. For example, the section on “Ratios andWeber fraction” argues that “these curves align well with the trend predicted by Weber’s law”, but does not explain how the experimental data would present if the alternative hypothesis (pairing-based strategy) was being used. What would the pairing-based strategy look like in Figure 6 right? Are there not significance tests that could be used to more carefully quantify the level of support for the two alternative strategies?
- The experiments seem very similar to Wu et al. 2018, which is considered to be prior work under the ICLR guidelines. While this paper is acknowledged in the related work, it would be helpful to expand further on the relationship between these works, so the originality and contribution of this paper can be better evaluated.
- In some ways it is not that surprising that the CNN more easily learns an approximate number system rather than a pairing-based algorithm, as the later would presumably need to learn a different convolutional filter for every possible spatial arrangement of the pairs (which would be very sample inefficient). Therefore, it might be interesting to consider, are there any circumstances under which the CNN would learn a pairing based algorithm? For example, what if the spatial configuration of the pairs was simplified, so they were always side-by-side at a fixed distance? If pairing-based algorithms emerged under simplified scenarios, this might have implications for the design of CNN filters (if we want models that are capable of learning these types of functions).

Summary:
I regard this as a good paper, with a couple of weakness that could be addressed as indicated.

---

> ### Author Response · Authors · 2018-11-16
> **Authors' response**
>
> Many thanks for the valuable feedback! We uploaded a revised version of the paper, and in the following address the weaknesses you pointed out:
>
> - Due to space constraints, we have to refer to Pietroski et al.'s work for more elaborate reasoning regarding the cognitive implications. We think their experiments are supposed to give strong indication for the ANS as a likely explanation of human behavior, and thus in our work for the FiLM model, without ultimately ruling out the pairing-based strategy (which is probably impossible via experiments evaluating extrinsic behavior only). We are not aware of what the curves for the pairing-based strategy in figure 6 would look like, but there is definitely evidence for Weber's Law in other approximate systems (where pairing-based strategies are no alternative), thus suggesting that similar mechanisms are at work here. We added a footnote on this to section 2.4.
>
> - We added a sentence to section 5 to clarify the differences (they focus on subitizing while we focus on ANS, and their experiments follow a different methodology with specifically designed data used for training and not just evaluation).
>
> - We actually consider the pairing-based strategy as more likely to be learned. Why? You're right that the convolutions need to learn to handle all possible spatial arrangements, but we think that this is the case for both the pairing- and the cardinality-based strategy, while the latter in addition needs to learn a presumably (our intuition) more complex aggregation mechanism of the locally computed results. Anyway, we added a few sentences to the end of section 2.4 discussing our intuition in some more detail to address this point.

---

### Meta-Review · Area_Chair1 · 2018-12-14

**Confidence:** 4
**Recommendation:** Reject

**Metareview:**

The paper studies an narrowly focused but interesting problem -- if the Visual Question answering model “FILM” from Perez et al (2018) is able to decide if “most” of the objects have a certain attribute or color. While the work itself is appreciate by the reviewers, concerns remain about the conclusion being limited in scope due to the synthetic nature of the data, and the analysis fairly narrow (a single model with a single very specific task). We encourage the authors to use reviewer feedback to make the manuscript stronger for a future deadline.